# Biopsychosocial Factors That Influence the Purpose in Life among Working Adults and Retirees

**DOI:** 10.3390/ijerph20085456

**Published:** 2023-04-10

**Authors:** Anabela Coelho, Manuel Lopes, Marta Barata, Sofia Sousa, Margarida Goes, Florbela Bia, Ana Dias, Ana João, Leonel Lusquinhos, Henrique Oliveira, Tânia Gaspar

**Affiliations:** 1Escola Superior de Enfermagem São João de Deus, Universidade de Évora, 7000-811 Évora, Portugal; 2Comprehensive Health Research Centre (CHRC), Universidade de Évora, 7004-516 Évora, Portugal; 3H&TRC-Health & Technology Research Center, ESTeSL-Escola Superior de Tecnologia da Saúde, Instituto Politécnico de Lisboa, 1549-020 Lisboa, Portugal; 4Global Health and Tropical Medicine, Instituto de Higiene e Medicina Tropical, Universidade NOVA de Lisboa, 1099-085 Lisboa, Portugal; 5Aventura Social Associação, Universidade Lusófona/SPIC, 1749-024 Lisboa, Portugal; 6Center for Interdisciplinary Heath Reseach (CIIS), Universidade Católica Portuguesa, 1649-023 Lisboa, Portugal; 7Instituto de Telecomunicações (IT-Lisboa), 1049-001 Lisboa, Portugal; 8Instituto Politécnico de Beja, 1049-001 Beja, Portugal; 9Hei-Lab, ISAMB, Universidade Lusófona, 1749-024 Lisboa, Portugal

**Keywords:** aged, aging, purpose in life, retirement, social adjustment

## Abstract

This study aimed to identify and characterize biopsychosocial factors that impact the purpose in life (PIL) among adults that are working or already retired. This cross-sectional study includes a sample of 1330 participants, of whom 62.2% were female, with ages ranging from 55 and 84 years, with a mean of 61.93 years and a standard deviation of 7.65. Results suggest that the education level, stress, spirituality (religion) and optimism, social support from friends, and quality of life related to physical health seem to contribute positively to the PIL for both groups. However, some variables such as age, marital status and environmental quality of life help explain the PIL of retired people and the quality of life related to social support helps explain the PIL of working adults. Overall, the reported findings suggest that the purpose in life is strongly related to physical, psychological, social and environmental health factors. It is highlighted that working adults and retired people have their purpose in life related to similar factors and others specific to each life stage, suggesting the need for crucial interventions to promote a healthier and more positive aging process.

## 1. Introduction

The purpose in life is a complex psychological construct that reflects the fact that an individual experiences his or her life in an organized way, oriented and motivated by objectives. This construct includes the individual’s life goals and their desire and determination to achieve them, encompassing the existence of a sense of direction and purpose [1,2,3,4]. The purpose in life promotes an intrinsic motivation to adopt healthy behaviors. It has been associated with positive health outcomes in older individuals [1,3,4,5,6], namely a lower risk of burnout [7], a lower risk of physical symptoms [8], the development of diseases, chronic conditions, disabilities, cognitive decline and mortality [3,4,5,9,10,11,12,13,14,15], as well as an increase in the positive effect [8,15] and greater satisfaction with life and the quality of life [4,15,16,17].

Several studies report factors that have significant association with a stronger purpose in life, namely socio-demographic factors such as having a high educational level [1,9,18,19], being married [1,18,19] and earning a high income [1,19]. Other factors are also reported, such as health and well-being [1,4,18,20], physical activity [19,21], health literacy [4], optimism [21], resilience [1,4,20], social support [4,19], social integration (family and social relationships, frequent contact with friends, social participation [1,19,20,21,22,23] and engaging in prosocial behaviors such as volunteering [21]), having broad perceptions of their time horizons/future [24], and faith and spirituality [1,4]. On the other hand according to the scientific literature, there are also several factors that are associated with a weaker sense of purpose in life, namely increasing age [2,4,14,18,19,25], a low education level [14], living alone [4,22], unemployment [2,22], financial stress [4], existence of a life-threatening or chronic illness [2,19,21,26], taking medication [19], losing a spouse [2], a reduced number of friendship relationships [22], hopelessness [21,22], loneliness [19,21,22], discrimination [19], the presence of anxious or depressive symptomatology [19,21,22], and the impact and fear of COVID-19 [24,27].

Increasing age has been found in some studies to negatively impact the purpose in life due to a loss of social roles as a consequence of the aging process [2,23]. Retirement is one of the social changes associated with this stage of life [2,23,28,29] and is associated with the end of working time and a fixed paycheck [29].

In Portugal, in December 2020, there were 2,969,728 Social Security pensioners, 2,070,387 age pensions, 178,577 disability pensions, 720,764 survivor pensions (benefits paid to a dependent of the person who has died) [30], and 648,647 “Caixa Geral de Aposentações” pensions (pension for public servants) [31]. Currently in Portugal, the retirement age due to old age is 66 years and four months [32], and it will remain the same in 2024 [33].

Retirement can significantly impact individuals’ lives, especially those who have been professionally active for most of their lives [29]. Work provides structure, goals, and a sense of identity, so retirement can be considered as a stage that leads to significant changes in individual’s perception of himself or herself and his or her life, by leading to a loss of structure, goals and roles to play [25]. In several cases, this event can be experienced as a loss and lead individuals to experience unpleasant feelings and difficulties in organizing their new life, which is exacerbated when this transition occurs abruptly [29].

Retirement can be associated with a diminished sense of purpose in life [2,22], leading individuals to feel lost and aimless [25] and to seek purpose in other nonworking roles [34]. Despite these findings, other studies found that individuals who had a low socioeconomic status and experienced dissatisfaction with their jobs showed a better sense of purpose in life after retirement, suggesting that retirement may also be an opportunity to experience a renewed purpose in life [25].

Developing strategies that empower working adults to prepare themselves for their retirement process [29,34] and identifying factors that influence individuals’ purpose in life is important [1].

This study aims to understand and characterize the biopsychosocial factors that influence the purpose in life (PIL) of adults that are working or retired. It is our purpose to provide an insight for relevant health and social policy development in Portugal and internationally.

## 2. Materials and Methods

### 2.1. Study Design

The study design was cross-sectional.

A structured interview methodology was adopted using the instruments mentioned in Section 2.3. All the instruments were made available in paper or digital (web-based) versions.

### 2.2. Participants

The sample included 1330 Portuguese participants who consented to participate in the study. The inclusion criteria were: (i) participants working or retired with or without a professional activity; (ii) participants interested in participating in the study. The exclusion criterion was having dementia diagnosed by a medical doctor, because abstract concepts, presented in the instruments, were considered too hard for people with dementia.

The convenience sample is not representative of the population; however, the large sample of 1330 participants includes people from different regions of Portugal and represents sociodemographic characteristics that are illustrative of the population under study.

### 2.3. Instruments

Sociodemographic data (age, gender, marital status, education level, and employment status) were obtained using a short survey and the following instruments were used:Psychosocial factors at work (PFW) were assessed through the Portuguese short version of Copenhagen Psychosocial Questionnaire, COPSOQ [35]. The main objective of COPSOQ is to evaluate psychosocial factors at work [36]. It consists of 40 items assessed on a 5-point Likert-type scale (1—Never/hardly ever; 2—Rarely; 3—Sometimes; 4—Often; 5—Always; or 1—Nothing/almost nothing; 2—A little; 3—Moderately; 4—Very; 5—Extremely), and has a Cronbach α of 0.89. The items on offensive behaviors were deleted; thus, the final version of the instrument used included 36 items measuring the following dimensions: (1) demands at work (quantitative demands, work pace, emotional demands); (2) work organization and job content (influence, development opportunities, purpose in work, commitment to the workplace); (3) interpersonal relations and leadership (predictability, recognition, role clarity, quality of leadership, social support from the supervisor); (4) work–individual interfaces (job satisfaction, work–family conflict); (5) values at the workplace level (trust regarding management, justice); (6) health and well-being (general health perceptions, burnout, stress). COPSOQ II follows a multidimensional concept, capturing the general needs involved within the “work stress” concept. If participants were retired, they were asked to report how long they had a professional activity.The purpose in life (PIL) was assessed through the PIL test, to assess the degree of existential emptiness, and one’s level of self-realization and appreciation of the purpose in life [37]. The instrument was translated and adapted for the Portuguese population (older adults), comprising 20 items, measured on a 7-point Likert-type scale, assessing: the meaning in life, satisfaction with life itself, freedom, fear of death, suicidal ideas, and if life is worthwhile, with a Cronbach α of 0.88 [38,39].The quality of life (QoL) was assessed through the Portuguese validated version of the World Health Organization Quality of Life-Brief (WHOQOL-BREF) instrument, with a Cronbach α of 0.92 [40]. It is a generic, multidimensional, and multicultural measure, for a subjective assessment of the quality of life, and can be used in a wide spectrum of psychological and physical disorders, as well as with healthy individuals. It consists of 26 items and integrates four domains of the quality of life: physical, psychological, social relations and environment. Each of these domains are characterized by between “3” and “8” characteristics representing that domain. Domain scores are summed to create a global score of the overall quality of life. This cross-cultural instrument can be applied to individuals living within different contexts, and capture their own views of their well-being.Social support satisfaction (SSS) was assessed using the scale ESS “Escala de satisfação com o suporte social”, with a Cronbach α of 0.85 [41]. It assesses perceived social support, considered a fundamental dimension in the cognitive and emotional processes linked to an individual’s well-being and the quality of life. It assesses satisfaction with friends/acquaintances (SF/A), intimacy (IN), satisfaction with family (SF), and social activities (SA). The instrument, with a total of 15 items (such as “My spiritual/religious beliefs give meaning to my life”; “My faith and beliefs give me strength in difficult times”; “I feel that my life has changed for the better”, etc.) is measured on a 5-point Likert scale ranging from “totally agree”, “agree mostly,” “neither agree nor disagree”, “disagree mostly”, to “totally disagree”.Spirituality, a sociocultural and historical construct, was assessed using the scale “Escala de avaliação da espiritualidade em contextos de saúde”, with a Cronbach α of 0.74, [42] comprising two dimensions: the religious dimension of hope (associated with a relationship with the transcendent) and optimism as an existentialist dimension that fits into the sense of beliefs related to the meaning of life.Cognitive style inventory (CSI) was the instrument employed to assess the cognitive response patterns associated with activities such as thinking, problem-solving and decision-making. The 34-item questionnaire identifies the cognitive style: “systematic style” and “intuitive style”, with a Cronbach α of 0.93 [43] in order to apply preferences and regular cognitive patterns in unconscious or deliberate responses. For example, the systematic style is illustrated by the statement “I usually do an orderly search for additional information and choose sources carefully” and the intuitive style by “I usually rely on ‘hunches’, instincts or other non-verbal cues to help me solve a problem”.

### 2.4. Data Acquisition Procedures

Instruments were administered by professionals from various organizations that decided to join the project and help collect data at different locations in Portugal, such as universities, unions, companies, municipalities, day-care centers, and non-governmental organizations that work with adults. Thus, 19 institutions/organizations were involved in the data collection task and were recruited through personal contacts. Information about study objectives was provided in full to respondents and/or their families, and they were informed about confidentiality and anonymity of the data. An informed consent form was delivered to the respondent or his/her family before each interview. No incentives were given to participants.

Data was collected through an online platform or in a paper-and-pencil version when participants were less technologically literate.

### 2.5. Statistical Procedures

Sociodemographic information was descriptively analyzed.

After invoking the Central Limit Theorem (large N), Pearson’s correlation coefficient was the measure of association used to characterize the strength of linear associations between the studied variables. Multilinear linear regression (using the ENTER method and PIL as the dependent variable) was used to obtain a parsimonious model to identify predictors of the purpose in life total score according to two scenarios: (i) working adults; (ii) retirees. The model’s assumptions were analyzed: normal distribution, homogeneity, and error independence. The first two assumptions were validated graphically, and the assumption of independence of errors was validated using the Durbin–Watson statistic (d = 1.59) [44]. The variance inflation factor (VIF) was used to identify multicollinearity among the variables. Outliers were examined (data with a studentized residual, in absolute value, more significant than 1.96). Although some outlier cases were found, they were kept in the models, since they did not significantly affect their goodness of fit.

All statistical analyses were performed with the software IBM SPSS Statistics for Windows v.27 (IBM Corp., Armonk, NY, USA). Type I error probabilities (α) of 0.05 and 0.10 were considered for all analyses.

### 2.6. Ethical Considerations

Ethics approval for this study was granted by Lisbon and Tagus Valley Regional Health Administration (“Administração Regional de Saúde de Lisboa e Vale do Tejo”, ARSLVT), reference number ARSLVT/Health Ministrypro.023/CES/INV/2014 [45].

## 3. Results

### 3.1. Sociodemographic and General Characteristics

From the 1352 responses obtained, 1330 were considered valid, with no missing values, of which 62% were obtained in paper format and 38% were obtained digitally.

Table 1 presents the sociodemographic results. More than half of the respondents were female (62.2%) and 37.1% were over 65 years old. Most respondents lived with someone (64.8%), and about 63.8% had completed mandatory schooling. The proportions of working people and retirees were similar (approximately 50%). The Purpose in Life scale average total score was 5.17 (SD = 0.74). No multicollinearity among the variables was identified.

### 3.2. Sample Characterization Concerning Occupational Status

More than half of the respondents were women, either working or retired. Concerning marital status, the proportion of individuals living with someone is always higher than those living alone, whether working or retired. For educational level, the proportion of retired individuals who have completed higher education (28.0%) is lower than that of working individuals who have completed higher education (42.4%) (Table 2).

### 3.3. Variables Scoring

Retirees presented higher levels of stress and burnout, and lower levels of spirituality (religion) and optimism than those working, and showed lower values statistically significant in all dimensions of the purpose in life than those who were working. Regarding social support, statistically significant differences were found concerning intimacy and family, with retired individuals reporting lower levels of intimacy and higher levels of social support from family than those who were working. Finally, for the quality of life domains, statistically significant differences were found, with retirees showing lower values in these dimensions than those who were working (Table 3).

### 3.4. Associations among Variabels (Working People)

Measures of associations between studied variables were obtained for working adults based on Pearson’s correlation coefficient, whose results are listed in Table 4.

The results suggest that the purpose in life total score (PIL) is associated:Moderately with “Psychological health”, “Physical health”, “Social relationships” and “Environment” domains of the quality of life (r values equal to 0.66, 0.48, 0.42 and 0.41, respectively), with “Spirituality (religion) and optimism (r = 0.53)”, and with “Intimacy” domain of social support (r = 0.53);Poorly with “Friends”, “Family“ and “Activities” domains of social support (r values equal to 0.37, 0.31 and 0.23, respectively), with “Stress and burnout” (r = 0.34), and with “Systematic cognitive style” (r = 0.25).

An overall analysis of the results listed in Table 4 suggests that as the score of all variables increases, the purpose in life (PIL) total score also increases since all measures of association are positive.

### 3.5. Associations among Variables (Retirees)

As for the working individuals (Section 3.4), measures of association between studied variables were also obtained for retirees, based on Pearson’s correlation coefficient. The respective results, listed in Table 5, suggest that the purpose in life (PIL) total score is associated:

Moderately with “Psychological health”, “Physical health”, “Environment” and “Social relationships” domains of the quality of life (r values equal to 0.66, 0.52, 0.45, and 0.41, respectively), with the “Intimacy” domain of social support (r = 0.53), with “Spirituality (religion) and optimism” (r = 0.47) and with “Stress and burnout” (r = 0.46);Poorly with “Friends”, “Family “ and “Activities” domains of social support (r values equal to 0.38, 0.37 and 0.24, respectively), and with “Systematic cognitive style” (r = 0.36).

An overall analysis of the results listed in Table 5 suggests that as the score of all variables increases, the purpose in life (PIL) total score also increases, since all measures of association are positive.

### 3.6. Predictors of the Purpose in Life Total Score (Working Adults)

The first multiple linear regression model (using the ENTER method) was designed using the purpose in life total score as a dependent variable and those listed in Table 6 as independent variables regarding working adults. This model was statistically significant $\left(F\left(16,315\right)=24.966;\;p<0.001;{\;R}_{a}^{2}=0.537\right)$, of which the results are listed in Table 6. This model suggests that the “Education level” $\left(B=0.16;\;t=2.87;\;p<0.01\right)$, “Stress and burnout” $\left(B=0.22;\;t=4.78;\;p<0.001\right)$, “Spirituality (religion) and optimism” $\left(B=0.20;\;t=2.87;\;p<0.01\right)$ and “Friends” domains of social support, and “Physical health” $\left(B=0.51;\;t=6.52;\;p<0.001\right)$ and “Social relationships” $\left(B=-0.13;\;t=-2.13;\;p<0.05\right)$ domains of the quality of life appear to be statistically significant predictors of the working respondents’ purpose in life total score.

### 3.7. Predictors of the Purpose in Life Total Score (Retirees)

Finally, the second multiple linear regression model (using the ENTER method) was designed using the purpose in life total score as a dependent variable and those listed in Table 7 as independent variables relating to retirees. This model was statistically significant (F(16,275) = 28.982; *p* < 0.001; $R_{a}^{2}$ = 0.606, of which the results are listed in Table 7. This model suggests that “Age” B = −0.02; t = −2.75; *p* < 0.01, “Marital status” B = 0.15; t = 2.36; *p* < 0.05, “Educational level” B = 0.15; t = 2.35; *p* < 0.05 “Stress and burnout” B = 0.18; t = 4.02; *p* < 0.001, “Spirituality (religion) and optimism” B = 0.26; t = 3.48; *p* < 0.01 and “Friends” domains of social support B = 0.16; t = 3.01; *p* < 0.01, and “Physical health” B = 0.46; t = 6.38; *p* < 0.001 and “Environment” domains of the quality of life appear to be statistically significant predictors of the respondents retirees’ purpose in life total score.

## 4. Discussion

This study aimed to identify and characterize the biopsychosocial factors that impact the purpose in life (PIL) among adults that are working or already retired.

The results suggest that for retired and working people, education level, stress, spirituality (religion) and optimism, social support from friends, and the quality of life related to physical health seem to contribute positively to PIL, as already supported by several studies [1,4,21].

What distinguishes the groups is that age, marital status and environmental quality of life help explain the PIL of retired people, and the quality of life related to social support explains the PIL of working adults.

### 4.1. Age

It is worth highlighting that, in this study, age is a negative predictor of PIL, which is somehow aligned with the available evidence which indicates a deterioration of the purpose with increasing age [2,19,22,46,47], even in participants still working [46]. Those who still work during the aging period have a lesser sense of purpose and meaning in life once their activity may be due to financial needs more than a source of meaning or purpose in life [46].

Although aging can involve a decrease in the meaning and purpose in life due to different age-related losses, even though our study does not reveal this relationship, we strongly believe, according to other several studies, that the quality social relationships help older adults build and integrate supportive social connections that can serve as a tool to maintain their focus on the new purpose in life [1,4,19].

Older age is a sign of more wisdom, experience and achievement of goals in life, so it is of utmost relevance, considering our findings, that we could help older adults keep a raised sense of purpose and meaning in life, for example, through more active and engaged aging [48].

### 4.2. Marital Status and Social Relationships

Aging and, in particular, the transition to retirement can represent a life shock to older adults, because free time increases enormously, becoming necessary to organize activity patterns and the time spent on them [49].

Regarding retirees, the results presented in this paper suggest that marital status seems to be a positive predictor of the PIL. It is proven that the absence of intimate relationships, the loss of a spouse [2], and loneliness are critical and negatively related to the PIL in older people [19,21,22].

The retired participants are older, so more often, age may weigh on their purpose in life and process of self-acceptance. They may be widowed and more lonely, and, on the other hand, the lack of physical and cognitive skills, technological skills, and socioeconomic limitations can limit the social circles and sense of belonging of the elderly [46,47].

In contrast to marital status, social relationships, as a QoL domain, seem to be a negative predictor, statistically significant in retirees. One reason for this finding may be related to Portugal’s low-income levels and low retirement incomes, which may negatively affect the purpose in life, especially for older adults of a lower socioeconomic status [30,31]. Thus, there is some evidence that not all retired individuals are prepared to build a new identity and outline their life projects, particularly if they lack social or financial support [4,13].

The decrease in social relationships during retirement may constrain the development of leisure activities, causing isolation for older adults, thus worsening the aging process and increasing the incidence of chronic diseases [50]. These biological and sociodemographic factors highlight that, for example, maintaining social relationships positively influences work achievement, social relationships and personal fulfillment, and may determine a greater PIL [1].

However all these results are aligned with those reported by other researchers, who revealed that age, marital status and social relationships are significant predictors of the PIL in retired populations [1,9,18,19].

Our study reveals that social support explains the PIL of working adults. The relation between the PIL and social support of the adults have been documented in other studies [4,19], as well as social activities/relationships with family, friends and social participation [1,19,20,21,22,23]. For several researchers, being alone is related to a low sense of PIL once gratifying relationships will support adults across their life challenges [6,19].

### 4.3. Environment

Several studies have shown that having a purpose and meaning in life empowers the execution of activities in older adults, making him/her feel well and in control of decisions about his/her life, providing a sense of self-realization and motivational power that helps him/her have a more optimistic outlook on his/her life experiences [46,51].

Effects of retirement demonstrated no significant change in well-being scores pre- and post-retirement [52], because it differs both between and within individuals over time [53,54,55,56].

In this study, the environmental domain of the quality of life is a negative predictor of the PIL once older adults may face more environmental barriers that affect their autonomy.

Several studies have shown that environmental barriers decrease autonomy, activities and opportunities of the elderly, which may affect their purpose in life [2,4,14,18,19,25].

### 4.4. Stress and Bournout

Another important finding in this study is related to stress and burnout, which is higher in retirees than in working individuals. As is known, health and well-being are not only determined by our genetics and personal characteristics, but also by physical and social environment, and the environment in which people live (epigenetics). In some studies, it has been proven that in retirement, life is more stressful, costs more than planned and is lonely [57]. Furthermore, reaching retirement age can presage the end of a productive and active life, subsequently evoking anxiety about being old and expendable. All these elements play an essential role in emotional and cognitive health. Poor social interaction, for example, can be associated with the deterioration of cognitive functions and depression. Therefore, older adults should have access to healthy, diverse and dynamic environments, where they can explore their potential until the end of their life [58].

The purpose in life is associated with the fulfillment of proposed personal objectives. In older adults, valuing those aspects that were part of their fulfillment and growth during their lifetime becomes relevant in how they perceive their day-to-day reality, not being limited only to the recall of goals achieved being crucial, but also having a purpose in life that facilitates the daily events to be evaluated positively [26].

In this study, optimism and spirituality are positively associated with the PIL, as also shown by other studies [1,4]; therefore, we must emphasize the importance of considering these dimensions in a global and holistic health strategy once, as proved by some studies, older adults can improve their commitment to care due to motivation and commitment to improving their health, as well as contribute to improving their levels of physical and psychological well-being, generating feelings of fulfilment and satisfaction with life [59].

## 5. Limitations, Strengths and Future Research

The study presents some limitations related to the study design. The cross-sectional design of our study limits our ability to determine causality between the PIL and these correlates. The present study reveals a statistical analysis of somewhat oversimplified data because it is an exploratory study, with very rich and original data. In future, it is intended to move to a more specific and deeper analysis; however, our findings are considered an important contribution.

The variability in the purpose in life can be shaped and modified through appropriate practices and policies that mitigate the negative effects of the biopsychosocial factors identified in this study. In other words, it is not clear that the meaning of the purpose in life is a stable concept between different groups of people. Thus, further qualitative research is needed to define what it the purpose in life means, in order to better understand whether the groups, namely working adults and retirees, are answering differently to these questionnaires or if, in fact, they see the purpose in life through a different construct.

The need for intervention strategies that enhance the PIL for the study population, considering the biopsychosocial factors identified in this study, is emerging. Such a scenario encourages further research and controlled interventions, in future terms, building a greater level of empirical evidence on the topic.

The findings and conclusions drawn in this article will contribute to new paths of research, and eventually review or redesign some of the health and social policies in Portugal, aiming at more consistent, participatory and transparent policies and practices.

## 6. Conclusions

The present study highlights that psychosocial factors are determinants of the purpose in life in adults, which may be impacted by adversities and losses associated with aging and retirement, particularly in older adults. It is recognized that not all older adults have the same ability to choose their life path and life purpose. Thus, further interventions should be planned regarding this multidimensional challenge, and it should be on the political agenda to determine strategies that prevent loneliness and promote a healthy life throughout the aging period.

## Figures and Tables

**Table 1 ijerph-20-05456-t001:** Sample sociodemographic and general characteristics (descriptive statistics).

Variable	N (%)	Min–Max	Mean (SD)
Sex: Male Female	481 (37.8) 793 (62.2)	- - -	- - -
Age: ≤64 years old ≥65 years old	835 (62.9) 493 (37.1)	55–84 - -	61.93 (7.65) - -
Marital status: Living with someone Living alone	846 (64.8) 460 (35.2)	- - -	- - -
Educational level: Up to mandatory schooling Higher education	821 (63.8) 465 (36.2)	- - -	- - -
Occupational Status: Working Retired	601 (50.3) 594 (49.7)	- - -	- - -
Chronic illness or health condition with impact on day-to-day life: No Yes	840 (65.6) 440 (34.4)	- - -	- - -
Takes medication for illness or physical condition: No Yes	647 (58.3) 462 (41.7)	- - -	- - -
Scoring “Purpose in life”: Happiness Meaning Willingness to live Total	1203 1175 1186 1083	1.50–7.00 1.43–7.00 2.14–7.00 2.30–6.70	5.17 (1.04) 5.3 (0.98) 5.47 (0.85) 5.17 (0.74)

**Table 2 ijerph-20-05456-t002:** Sample characteristics concerning occupational status (descriptive statistics).

Variable	Working	Retired
Age:		
Min.	55	49
Max.	72	84
Avg	56.29	67.33
SD	5.18	5.55
Sex: Male Female	201 (34.8) 377 (65.2)	232 (40.9) 335 (59.1)
Marital status: Living with someone Living alone	418 (69.9) 180 (30.1)	346 (60.4) 227 (39.6)
Educational level: Up to mandatory schooling Higher education	337 (57.6) 248 (42.4)	414 (72.0) 161 (28.0)
Chronic illness or health condition with impact on day-to-day life: No Yes	417 (71.8) 164 (28.2)	337 (58.8) 236 (41.2)

**Table 3 ijerph-20-05456-t003:** Descriptive and F statistics of the studied variables regarding retirees and those who were working.

Variable	Working		Retired		
	N	Mean (SD)	N	Mean (SD)	F
Stress and burnout	578	12.58 (3.15)	549	13.02 (3.14)	5.52 *
Spirituality (religion) and optimism	565	2.73 (0.73)	530	2.58 (0.72)	10.48 **
Systematic cognitive style	499	3.54 (0.42)	469	3059 (0.41)	3.71
Purpose in life: Happiness Meaning Willingness to live Total	558 542 549 508	5.31 (0.99) 5.34 (0.94) 5.54 (0.85) 5.25 (0.72)	531 523 523 474	4.98 (1.07) 5.21 (1.02) 5.36 (0.81) 5.05 (0.75)	28.24 *** 4.93 * 12.92 *** 18.67 ***
Social support: Friends Intimacy Family Activities Total	561 568 571 571 541	3.58 (0.63) 3.63 (0.82) 3.78 (0.82) 2.91 (0.87) 3.51 (0.56)	537 542 556 542 504	3.61 (0.66) 3.50 (0.81) 3.88 (0.87) 2.92 (0.90) 3.51 (0.58)	0.57 6.78 ** 4.31 * 0.03 0.04
Quality of life: Physical health Psychological health Social relationships Environment Total	547 551 566 532 575	3.95 (0.65) 3.92 (0.58) 3.93 (0.61) 3.46 (0.59) 3.73 (0.67)	527 534 513 533 565	3.76 (0.70) 3.85 (0.61) 3.78 (0.63) 3.49 (0.57) 3.58 (0.71)	21.07 *** 3.92 * 17.21 *** 0.55 13.98 ***

* *p* < 0.05; ** *p* < 0.01; *** *p* < 0.001.

**Table 4 ijerph-20-05456-t004:** Pearson’s correlation coefficients between studied variables (working people).

Variable	SB	SO	SCS	PIL	SS_fr_	SS_i_	SS_fa_	SS_a_	QoL_ph_	QoL_p_	QoL_sr_
Stress and burnout (SB)	-	-	-	-	-	-	-	-	-	-	-
Spirituality and optimism (SO)	0.27 ***	-	-	-	-	-	-	-	-	-	-
Systematic cognitive style (SCS)	0.05	0.20 ***	-	-	-	-	-	-	-	-	-
Purpose in life (PIL)	0.34 ***	0.53 ***	0.25 ***	-	-	-	-	-	-	-	-
Social support: Friends: SS_fr_ Intimacy: SS_i_ Family: SS_fa_ Activities: SS_a_	0.17 *** 0.29 *** 0.26 *** 0.23 ***	0.37 *** 0.45 *** 0.26 *** 0.10 *	0.14 ** 0.08 0.14 ** 0.00	0.37 *** 0.53 *** 0.31 *** 0.23 ***	- 0.57 *** 0.41 *** 0.29 ***	- - 0.39 *** 0.33 ***	- - - 0.16 ***	- - - -	- - - -	- - - -	- - - -
Quality of life: Physical health: QoL_ph_ Psychological health: QoL_p_ Social relationships: QoL_sr_ Environment: QoL_e_	0.45 *** 0.47 *** 0.29 *** 0.37 ***	0.31 *** 0.50 *** 0.38 *** 0.40 ***	0.03 0.12 * 0.130 ** 0.04	0.48 *** 0.66 *** 0.42 *** 0.41 ***	0.30 *** 0.36 *** 0.45 *** 0.24 ***	0.43 *** 0.50 *** 0.52 *** 0.39 ***	0.26 *** 0.32 *** 0.36 *** 0.23 ***	0.23 *** 0.18 *** 0.19 *** 0.30 ***	0.68 *** 0.47 *** 0.64 ***	- - 0.58 *** 0.56 ***	- - - 0.34 ***

* *p* < 0.05; ** *p* < 0.01; *** *p* < 0.001.

**Table 5 ijerph-20-05456-t005:** Pearson’s correlation coefficients between studied variables (retirees).

Variable	SB	SO	SCS	PIL	SS_fr_	SS_i_	SS_fa_	SS_a_	QoL_ph_	QoL_p_	QoL_sr_
Stress and burnout (SB)	-	-	-	-	-	-	-	-	-	-	-
Spirituality and optimism (SO)	0.22 ***	-	-	-	-	-	-	-	-	-	-
Systematic cognitive style (SCS)	0.20 ***	0.23 ***	-	-	-	-	-	-	-	-	-
Purpose in life total (PIL)	0.46 ***	0.47 ***	0.36 ***	-	-	-	-	-	-	-	-
Social support: Friends: SS_fr_ Intimacy: SS_i_ Family: SS_fa_ Activities: SS_a_	0.33 *** 0.35 *** 0.27 *** 0.26 ***	0.31 *** 0.36 *** 0.24 *** 0.15 **	0.24 *** 0.19 *** 0.21 *** 0.09	0.38 *** 0.53 *** 0.37 *** 0.24 ***	- 0.56 *** 0.42 *** 0.32 ***	- - 0.48 *** 0.35 ***	- - - 0.10 *	- - - -	- - - -	- - - -	- - - -
Quality of life: Physical health: QoL_ph_ Psychological health: QoL_p_ Social relationships: QoL_sr_ Environment: QoL_e_	0.52 *** 0.53 *** 0.40 *** 0.44 ***	0.20 *** 0.41 *** 0.32 *** 0.26 ***	0.22 *** 0.29 *** 0.23 *** 0.19 ***	0.52 *** 0.66 *** 0.41 *** 0.45 ***	0.34 *** 0.38 *** 0.54 *** 0.28 ***	0.33 *** 0.44 *** 0.48 *** 0.31 ***	0.20 *** 0.30 *** 0.38 *** 0.21 ***	0.23 *** 0.24 *** 0.22 *** 0.25 ***	- 0.66 *** 0.52 *** 0.67 ***	- - 0.52 *** 0.57 ***	- - - 0.50 ***

* *p* < 0.05; ** *p* < 0.01; *** *p* < 0.001.

**Table 6 ijerph-20-05456-t006:** Results of a multilinear regression model that was designed using the purpose in life total score as the dependent variable, involving only working people.

Independent Variable	*B* ^1^	S.E. ^2^	*β* ^3^	*t* ^4^
Age	−0.01	0.01	−0.05	−1.28
Sex	−0.10	0.05	−0.07	−1.83
Marital status	0.02	0.06	0.02	0.37
Educational level	0.16	0.06	0.12	2.87 **
Chronic disease or health condition	−0.03	0.06	−0.02	−0.44
Stress and burnout	0.22	0.05	0.24	4.78 ***
Spirituality and optimism	0.20	0.07	0.11	2.87 **
Systematic cognitive style	−0.06	0.05	−0.06	−1.11
Social support: Friends: SS_fr_ Intimacy: SS_i_ Family: SS_fa_ Activities: SS_a_	0.15 0.03 0.02 0.09	0.05 0.04 0.03 0.07	0.18 0.04 0.03 0.09	3.25 ** 0.91 0.74 1.39
Quality of life: Physical health: QoL_ph_ Psychological health: QoL_p_ Social relationships: QoL_sr_ Environment: QoL_e_	0.51 −0.00 −0.13 0.00	0.08 0.06 0.06 0.01	0.41 −0.00 −0.12 0.02	6.52 *** −0.04 −2.13 * 0.34

^1^ *B*: Non-standardized regression coefficient. ^2^ S.E.: Standard error. ^3^ Standardized regression co-efficient. ^4^ *t*-Student. * *p* < 0.05; ** *p* < 0.01; *** *p* < 0.001.

**Table 7 ijerph-20-05456-t007:** Results of a multilinear regression model that was designed using the purpose in life total score as the dependent variable, involving only retirees.

Independent Variable	*B* ^1^	S.E. ^2^	*β* ^3^	*t* ^4^
Age	−0.02	0.01	−0.11	−2.75 **
Sex	0.08	0.06	0.06	1.34
Marital status	0.15	0.06	0.10	2.36 *
Educational level	0.15	0.07	0.10	2.35 *
Chronic disease or health condition	−0.03	0.06	−0.02	−0.52
Stress and burnout	0.18	0.05	0.17	4.02 ***
Spirituality and optimism	0.26	0.07	0.14	3.48 **
Systematic cognitive style	0.09	0.06	0.08	1.51
Social support: Friends: SS_fr_ Intimacy: SS_i_ Family: SS_fa_ Activities: SS_a_	0.16 0.04 −0.01 0.09	0.05 0.04 0.04 0.07	0.17 0.05 −0.01 0.09	3.01 ** 1.02 −0.20 1.44
Quality of life: Physical health: QoL_ph_ Psychological health: QoL_p_ Social relationships: QoL_sr_ Environment: QoL_e_	0.46 −0.09 −0.02 0.02	0.07 0.06 0.07 0.01	0.38 −0.08 −0.02 0.11	6.38 *** −1.59 −0.33 2.15 *

^1^ *B*: Non-standardized regression coefficient. ^2^ S.E.: Standard error. ^3^ Standardized regression co-efficient. ^4^ *t*-Student. * *p* < 0.05; ** *p* < 0.01; *** *p* < 0.001.

## Data Availability

Data are unavailable due to privacy or ethical restrictions.

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
