# Peer review of "Biopsychosocial Factors That Influence the Purpose in Life among Working Adults and Retirees"

_ijerph, 2023, doi:10.3390/ijerph20085456_

Round 1

Reviewer 1 Report

I thank the authors for the opportunity to review this manuscript, examining how biological, psychological, and social factors are related to the meaning of life in the retirement process. To test these hypotheses, 1330 participants between the ages of 55 and 84 years completed self-report questionnaires. The results suggest that meaning of life is linked to physical, psychological, social, and environmental health factors. In this light, the strength of the study is that it has both theoretical and practical implications concerning the importance of interventions to foster healthy and positive retirement processes. I made a list of suggestions to try assist the authors improve their paper.

Introduction

The Introduction gives an in-depth summary of the relevant literature on meaning in life. In order to complete the review, I suggest that authors mention the findings showing that older employed adults reported lower meaning in life than those unemployed.

Greenblatt-Kimron, L., Kagan, M., & Zychlinski, E. (2022). Meaning in Life among Older Adults: An Integrative Model. International Journal of Environmental Research and Public Health19(24), 16762. https://doi.org/10.3390/ijerph192416762

Method

1.     I suggest the authors edit their paper due to typos throughout the paper, for example after each instrument replace ";" with a full stop.

2.     Please include Cronbach's alpha to all the relevant study instruments. This is crucial information that is missing.

3.     Please state how many items make up the instrument "Social support satisfaction" and each scale.

4.     Please give examples of the questions used for the instrument developed by Pinto & Pais-Ribeiro (2007) and correct the reference in the text.

5.     Please provide more information on the Cognitive Style Inventory.

Results

I suggest the authors to try not to repeat unnecessary information presented in the Tables by limiting the text to the most important outcomes.

Reviewer 2 Report

Overall:

Thank you for the opportunity to review this interesting paper. You have a great research question and data to enable you to respond to this question. However, the paper is marred by a number of errors (that may be due to translation), stylistic issues, definitional problems and some missing information that make it a difficult paper to read with ease. More targeted descriptions and judicious editing with an English speaker will help with the wordiness. This will help the overall readability of the paper.

One example of shortening information might be at section 2.6, Ethical considerations.  This paragraph could be shortened to: "Ethics approval for this study was granted by Lisbon and Tagus Valley Regional Health Administration (“Administração Regional de Saúde de Lisboa e Vale do Tejo” – ARSLVT ) reference number ARSLVT/Health Ministrypro.023/CES/INV/2014."  The other information you provide is standard practice and in general does not need to be included unless specifically requested by the journal or the committee granting ethical approval.  

A stylistic example is the two referencing styles you use. I far prefer one consistent style (unless there is a compelling reason to change). I give examples in sections of the review below.

I have also suggested some reworking and additional text in the Methods section.

With regards to definitions of key concepts. Throughout your paper you refer to the Meaning OF Life. In usual English expression the “Meaning OF Life” tends to be used as an existential question about the significance or reason for existence of the human race. Whereas, “Meaning IN Life”…reflects the feeling that one's existence has significance, purpose, and coherence (see Heintzelman and King, 2014). See: https://www.frontiersin.org/articles/10.3389/fpsyg.2020.601899/full   “Of” and “if” are both prepositions, but the change to ‘if’ would make a big difference to the flow of the article for me. I encourage your team to discuss this change.

Secondly, your article is entitled ‘Biopsychosocial factors that influence the meaning of life in the retirement process. A process is defined as a dynamic state where a series of actions or steps is taken to reach a certain end-point. I find it difficult with the cross-sectional design to justify how your paper describes the retirement process. What your paper does well is contrast how different life experiences, beliefs etc. etc. influence people’s perceptions of meaning of their lives in a range of pre-retirement and post retirement scenarios. At worst, ‘process’ is a little misleading and I encourage the team to consider an alternative title.

Sections of the paper:

Abstract:

Please consider changing "...the meaning OF life can be explained by age and education,...". Noting my concerns with expression above, these sentences could be changed to something like "the findings suggest that meaning IN life ..."

The comment regarding change ‘OF’ to ‘IN’ applies to the remainder of the paper.

Reading the abstract as a stand-alone text (as most people will do) I am not sure why you have a listing of explanatory variables (lines 26-28) followed by "the results also suggest..." (Lines 28-30) and some of the same variables are listed. If you could briefly explain if this second list of variables was derived from a different analysis or some other reason, it would lend greater clarity.

Introduction: p.2 line 48 "...positive affects". 'Affect' is usually considered a collective noun for a state of mood, therefore change to "positive affect".

P.2 Line 57: what is a ‘broad perception of the future’? Can you possibly expand?

P.2 Line 59: "a lower meaning of life" I find this expression awkward! You can have less of a sense of meaning in life, or less of a sense in purpose of life or attach less meaning to your existence. I suggest an alternative adjective. (Similarly with ‘raised meaning of lie” at line 417)

Continuing in this paragraph, you use qualifying words to describe ‘lower sense of meaning’ (eg increasing age, low education etc. The variables you list describing improved sense of meaning would also be better with adjectives or adverbs -  "...such as, for example, good health and well-being, engaging in physical activity... ". it gives clarity to the directionality!

P.2 Line 71Can you briefly explain what a pension by "survivors" is, and what 'CGA pensioners' are? I am not familiar with this term and I assume many other readers will also not know what this means.

P.2 line 85-86 you abruptly change referencing style. I feel it would be better to keep a consistent style. Consider changing to  " Despite this finding, a study examining XYZ found individuals of low socio-economic status and former job dissatisfaction.....renewed meaning in life [25]. "

P. 2 Line 93 The last paragraph could be condensed:  Eg. "This study aims to understand and characterize the biopsychosocial factors that influence the meaning of life in retirement and provide insight for relevant health and social policies development in Portugal and internationally."

The information about new paths of research would be better placed into a section on future research towards the end of the article.

Method:

2.1 Again, shorter is better! eg "Study design was cross sectional".

2.2 There is no information on how you recruited your sample. This is important to provide. I am curious - you have not stated an age in your inclusion criteria, though this would seem to be critical to a paper regarding retirement?

2.3 Keep it short: Eg "Sociodemographic data (age, gender, marital status, education level, and employment status) were obtained using a short survey.  You don't need to list the other factors and then repeat these in the description of instruments. 

Line 113,114. You don't need to reiterate "The COPSOQ’s main objective is to evaluate psychosocial factors at work, it is evident from the first line. In line 137 again, keep your referencing consistent.  You can say for eg "Quality of Life (QoL) was assessed through the Portuguese validated version of the World Health Organization Quality of Life – Brief (WHOQOL-BREF) instrument [40]."

Line 139 - you might want to list the four domains of the QoL scale.

Line 141 Consistency in referencing. eg "Social Support Satisfaction (SSS) was assessed using <name the scale here> [41]. The same comment applies for lines 148-149 and 153.  you don't need to say the name of who the scale was developed by - that's what the reference is for.

2.4 Data acquisition procedures. This is an important section that needs further work. You could consider combining it with an expanded section on sampling procedures.

As well as being explicit about how you recruited participants, you need to be clear about how you recruited the organisations involved. For example, state if this was systematic recruiting for sampling purposes, or a sample of convenience and so on.   You need to describe the settings and locations, were the organisations paid incentives etc.,etc., etc. Were the organisations systematic about how they recruited participants (did they use the same techniques and procedures etc as each other)?

I would suggest you move the methods you used for the survey and interviews to the study design section (paragraph beginning line 163). It helps to orient the reader.

The lines 165-167 belong in results.

Information on informed consent may be more appropriate in a new section on sampling.

It might be helpful to review a publication manual such as the APA Publication Manual - chapter on parts of a manuscript. Most of the information you already have, and a little more structure will help the reader. A key reference is:

American Psychological Association. (2020). Publication manual of the American Psychological Association 2020: the official guide to APA style (7th ed.). American Psychological Association.

Line 173. keep your expression short!  Eg. "Sociodemographic information was descriptively analysed." is really all that is needed.

Lines 175-183 (variables) please consider a table to present this information or in very small font as a legend under the results tables.

 Line 184  You can simply justify the assumption of normality ( also state you N rather than the more vague “many respondents”)

Line 195. Outliers were diagnosed? possibly 'examined' may be a better word?

Results: In the first paragraph you may consider pointing out only findings most worthy of remark. For example for a paper on meaning in life and retirement less than half the sample (37.1%) were over 65 and the minimum age in the range was 37 years, which does not seem to be valid to answer the research question to  “understand and characterize the biopsychosocial factors that influence the meaning of life in the process of retirement…” you should justify the age range.  Maybe the 37-year-old was considering early retirement? The information presented thus far in the paper does not really explain how your choice of sample will answer the research question.

Pages 7 and 8 - Lines 248, 264a typo in the word 'variables' (sic variabels)

Reporting of results. Please review your text.  Eg in 3.7 you write “…Meaning of Life total score as a dependent variable and those listed in Table 7 as dependent variables, concerning the entire sample.” The latter should be independent variables. Also see 3.8 and 3.9.

The contrast of results for working and non-working older people is useful, (especially given debate around retirement ages), however I would like to know what the ages of these two groups are and if there was a significant difference. I suggest adding this into Table 2. 

Discussion:

I found the discussion a tricky section to read. The research question should be re-iterated in the first paragraph. This orients the reader. Next you need to evaluate your findings and how they relate to your original research question AND where your results fit into, diverge from or add to f the current literature. Much of the information in the early paragraphs really needs to go into the introduction to ‘set the scene’ of current research and thinking about meaning in life.

Line 360: Please be realltycareful about clarity of expression. “The Meaning of Life is supposed to be cognitive in nature and usually formed early in life….”. Possibly a better way to express this would be “ meaning in life has been characterised as a cognitive construct formed early in life…” [ref]. “supposed” is a very imprecise term. As an international author don’t be embarrassed about asking international journals for English language assistance – many journal have this services

 In 3.3 and the discussion you refer to participants as being professionally active. Can you define this please? Is it, for example related to people in the traditional ‘professions’ – law, medicine, nursing etc or does to relate to any working role? Does it require continuation of income earning or does it include those retirees with volunteer jobs?  In line 439 you refer to ‘professional support’. I assume this meaning of the word refers to paid social support services?

Line 377 – stylistically, the text provided and referencing I think could be improved. For eg., this line reads “The study conducted by Henning, et al. [50] demonstrated that retirement does not significantly impact most people's well-being.” I find reference to “the study by Henning” not very helpful as it does not mean anything to me. What would be more helpful is to say (I am making this up) “A repeated measures survey of XXXX people in Germany demonstrated no significant change in well-being scores pre- and post-retirement”. This way I don’t have to look up the reference and it provides me with a reference point for your argument.

Line 388 – By here in the discussion, I find I am struggling to follow the narrative. It seems to jump around rather that follow a line of argument.

When I read “This scenario can result in difficulty finding a positive Meaning of Life and can lead to psychosocial or physical health problems. Even if they previously formed ideas and beliefs about their retirement, expectations may not match reality” this seems to be at odds with your statement on p 11 line 360 – 361where you mention meaning is established early in life.  You note meaning can be modified by personal experiences. By line 387 you have introduced the complexity that underlies this area of research; in line 390 you introduce a new concept of previously formed beliefs about retirement.

I feel this discussion needs much more structure to guide the reader through!

The final sentence of the discussion “The lack of thoughtfulness about the transition to retirement can lead individuals to develop practices and behaviours that do not promote QoL”  does not seem to arise from your study and introduces the concept of quality of life. While QoL was a measure in your assessment battery, the relationship of quality of life and meaning in life is not explored, nor is it clear to me that you explored thoughtfulness about retirement.

This paper needs a section on strengths, limitation, and future research and/or policy directions. 

I encourage you to revisit your key concepts and re develop the paper to focus on the research question, which should give your paper clarity and more direction.

Reviewer 3 Report

The manuscript, “Biopsychosocial Factors that Influence the Meaning of Life in the Retirement Process” presents a cross-sectional analysis of multiple indicators of self-reported meaning. Strengths of this manuscript is its inclusion of a large sample as well as the investigation of multiple predictive indicators within the same model. By including multiple predictors simultaneously, this paper stands to offer a more realistic view of what factors correspond with the experience of meaning – and to what relative degree they do so. While the topic being addressed in this manuscript could be of keen interest to readers of this special issue, I noted some limitations that obscured my sense of the contributions it is likely to make in its current form. To this end, I offer the following critiques:

1.     Authors may wish to reconsider their title as this cross-sectional title cannot support inferences regarding the “retirement process”. Without longitudinal or more intensive/dynamic data, answering process questions fall beyond the scope of what can be learned using cross-sectional analyses.

2.     The opening claim of the paper (lines 40-41) attempts to clarify what meaning is but ends up using the term “meaningful” later in the same sentence. This makes their definition of meaning circular and confusing. 

Perhaps related to this point, a particularly curious aspect of this paper is the use of the Purpose in Life test to assess meaning. It is striking that the Purpose in Life measure includes meaning as a subscale, but alongside two other subscales (happiness and willingness to live). First, this exposes a prima facie conflation of two concepts: purpose and meaning. Do the authors believe these two concepts are one in the same? Failure to reconcile this conflation leads to confusing interpretations throughout the paper, while ignoring important nuances in the psychological literature that has been built up around either construct. For example, there is much more experimental work in the meaning in life literature than in the purpose in life literature; and there is much more developmental work on purpose than on meaning. Moreover, given the title of the current paper and the focus of its introduction, I was expecting any one of the commonly used measures of meaning in life to have been deployed in the study. However, in the methods describing the instruments used in this paper, meaning doesn’t appear at all. Instead, only purpose in life is referenced. 

3.     Why doesn’t Table 2 report the age of participants in each group? I see age is reported for the overall sample but cannot seem to find the average age for the working and retired groups, respectively.

4.     Were the authors concerned about detecting so many positive bivariate correlations among predictors (multicollinearity) prior to running their multiple regression models? Everything assessed in these models appears positively and significantly associated with everything else at the bivariate level! For example, Table 4 (correlations for overall sample) suggests that stress and burnout and multiple indicators of social support were each positively associated with indicators of quality of life above .5, suggesting these variables have more in common with one another than what distinguishes them. Moreover, both psychological health and social support intimacy correspond with meaning in life at around .7. To this reader, analyzing these data using a structural equation modeling approach might afford greater protection against biased estimates.

5.     The Discussion frames the results as suggestive of meaning in life “during” the retirement process. But as stated earlier, there are no analyses presented here attempting to consider any processes at all. This leads to indefensible claims such as this one on lines 432-435: 

“Results of the present study also seem to suggest that retirees do not adjust quickly to their new life situation, as their affective and social support network decreases substantially…”

How are readers to interpret what ‘adjust quickly’ means when we do not know how long the retirees have been retired, what lead to retirement decisions, nor how close each member of the currently working group are to retirement? Have any of the currently working participants retired and started working again? Are there differences in meaning among those who have just retired recently vs. those who have been retired for some longer periods of time? Does self-reported meaning change as one moves from working to retirement status? Answers to these questions might set up a more process-oriented exploration. But as is, the current paper can merely describe differences in meaning between participants’ current work and retirement statuses, not processes.

6.     In the Discussion, the authors claim the negative association they report between age and meaning in life in both working and retired participants to be “aligned with the available evidence, which indicates a deterioration of Meaning of Life with increasing age and retirement”. Yet, ample evidence exists to the contrary – that is, that meaning can increase with age. See for example the following studies:

Krause, N., & Rainville, G. (2020). Age differences in meaning in life: Exploring the mediating role of social support. Archives of Gerontology and Geriatrics, 88, 104008.

King, L. A., & Hicks, J. A. (2021). The science of meaning in life. Annual review of psychology72, 561-584.

Baikeli, R., Li, D., Zhu, L., & Wang, Z. (2021). The relationship between time perspective and meaning in life across different age stages in adulthood. Personality and Individual Differences174, 110668.

Steger, M. F., Oishi, S., & Kashdan, T. B. (2009). Meaning in life across the life span: Levels and correlates of meaning in life from emerging adulthood to older adulthood. The journal of positive psychology4(1), 43-52.

Battersby, A., & Phillips, L. (2016). In the end it all makes sense: Meaning in life at either end of the adult lifespan. The International Journal of Aging and Human Development83(2), 184-204.

Notably, with respect to this point about the negative association between meaning and age, the authors cite two studies. Yet, both are reviews of the purpose in life construct; in fact, “meaning in life” is not focal in either paper cited, and is not mentioned at all in one of them. I note this in hopes that greater connection between the terms used and the literature reviewed can be achieved.  

Minor points:

a.     The phrase “the meaning of life” is used throughout but reads as odd – and unnecessarily grandiose – to me. The actual meaning of life is an existential commentary or perspective that isn’t under investigation here. Instead, the authors are merely documenting the extent to which participants reported feeling a “sense of meaning in their lives”.  The authors may wish to modify their wording.

b.     In the Abstract, “intimacy-related social support” is used at first, but then “social support related to intimacy” was used subsequently. If these are the same construct, then using the same descriptive phrasing would be helpful to readers.

Round 2

Reviewer 2 Report

Thanks for the opportunity to review this revision. The paper is much better but there is still a little way to go. below I give detailed comments that I hope will assist, especially in English expression.

Abstract

This abstract reads much better. I think avoiding ‘meaning of life’ was a wise choice.  It avoids the connotations of existential experience. It also relates to the references in a meaningful way.

In colloquial English, the article ‘the’ can often be dropped to create an easier flow of written language. I think you can do a minor edit and where not necessary remove THE from ‘purpose in life’.  eg the opening sentence The purpose in life can be more simply put as  “Purpose in life in a complex psychological construct…”. Likewise at line 43 Page 1 Line 50 page 2 and so on. See also line

Line 50 ‘Better purpose in life’ –I feel ‘better’ is the wrong adjective. It would be ‘stronger  purpose’ or a ‘stronger sense of purpose in life’.  I fell you mean magnitude of purpose here.

Likewise,  if you use stronger purpose, at line 58 you can have a weaker sense of purpose. I feel this now puts purpose onto a continuum of weaker to stronger and you nicely illustrate the variables that affects purpose.  

Line 64: delete the superfluous THE and impact will be plural: Increasing age negatively impacts purpose in life .. or “Increasing age has been found in some studies to negatively impact purpose in life due to loss of social roles as a consequence of the ageing process.”   The second part fo that sentence does not really make sense. … force the individual to adapt to and leave…  if the individual has adapted to changes brought about by the ageing process surely , they would not be relinquishing these roles?

Line 75 delete the THE. Just ‘Retirement can ….” Line 79 delete the THE in ‘the roles”

Line 80 perhaps consider changing “…experience unpleasant feelings and difficulties in organizing their new life, mainly when this transition occurs abruptly” to ““…experience unpleasant feelings and difficulties in organizing their new life, which is exacerbated when this transition occurs abruptly”.

Line 84 These findings (plural)

Lines 84-85 Change “low socioeconomic status and dissatisfied jobs” to “…low socioeconomic status and experienced dissatisfaction with their job…”

Line 88 delete the THE in the working adults.

End of introduction delete the additional text at the end “purpose in”

Overall,  the introduction reads much better! These mainly grammatical and stylistic change should further improve it.

Methods

 You are going to laugh. The opening sentence is one example where the article THE is quite appropriate. “The study design was cross sectional.” I am sorry English is such an absurd language!

2.2 I assume all participants CONSENTED. Change from ‘agreed’ to ‘consented’.

Try “A dementia diagnosis was the sole exclusion criterion.” It would be good to say why. Is for example dementia often diagnosed later in the disease meaning that abstract concepts presented in the instruments considered too hard for a people with dementia.  The reason I suggest this is that there is growing literature that questions the exclusion for older people with dementia in research.

Lines 107-109 are a little convoluted! Change “…the large sample of 1330 participants including people from different regions of the country can be consider relevant once it will represents sociodemographic characteristics that are quite illustrative of the population under study.

To “ …the large sample of 1330 participants includes people from different regions of the country and represents sociodemographic characteristics that are illustrative of the population under study.

2.3, Line 113 Instruments were administered or used, not considered.

Line 144-145

 Change from “Each of these domains is composed of facets of quality of life that summarize the particular domain of quality of life in which they are inserted.” To something more like “each of these domains are characterised by between <lower number> and <higher number> characteristics representing that domain.”

And at Line 146 and 147, perhaps “Domain scores are summed to create a global score of overall quality of  life” ( f this is what is done.)

Line 169 “in order to apply preferences, regular cognitive patterns and unconscious or deliberate responses” is it  regular cognitive pattern IN unconscious  or deliberate, or is really and AND?

Line 169 delete the ARE at the beginning of the sentence and check that the meaning is accurate. It may be better to say “For example, the systematic style is illustrated by the  statement “I usually do a orderly search… etc”

Line 175  “Instruments were administered (not applied) by professionals…” you need to say how these organisations were recruited. Did you advertise? Was it through personal contacts?  Were these 19 organisations networked in some way?

This detail is very important to understand. The data gathering statement is much better now ( line 181-185).

Line 192 delete the THE in “The multi linear regression”

Line 198. You can just say what you did and support  your methods with a reference. Eg change “…assumption of independence of errors was validated with the Durbin-Watson statistic (d=1.59), as suggested by the study [44]”. To  “…assumption of independence of errors was validated using the Durbin-Watson statistic (d=1.59) [44].

Line 199 “…to identify multicollinearity among the variables, which did not occur.” Which do. Not occur is a result and should move the results section.

Line 236 delete ‘te’ ( typo). ‘The’ is not needed here.

Please correct typos at lines 243-244 and formatting at Line 246

Line 241 delete ‘the’ in the working individuals.

Please correct formatting problems at lines 277-278 and 281-282.

Discussion

Great to read the summary of the purpose of the study in the first paragraph!

Please review this paragraph: “It is worth highlighting that, in this study, age is a negative predictor of PIL, which is somehow aligned with the available evidence, which indicates a deterioration of the purpose with increasing age [2,19,22, 46, 47] even though participants still working [46].  Those who still working etc…”

This does not make sense.  Can you please review for meaning and expression?

Perhaps consider writing “…even in participants are still working” the word ‘though’ is maybe not the best choice.

Be careful of statements like “ it is well known that…” this is supposed to be science! We can talk about assumptions – as assumptions – but we as scientists TEST assumptions!  I would review paragraph at 338 – 342. You make sound clinical sense in helping older people to remain active and socially engaged is but carefully relate this to your results.

Line 348 be careful again of assumptions. Yes, when retired people don’t meet and engage with colleagues, but they may meet and engage in other ways.  Make sure you relate these comments to your results and  contrast this with the current literature.

Line 381 change  “It is important to enhance that several studies…”  to perhaps just “ Several studies have shown…” ( use of the word ‘enhance is incorrect.)

Break paragraph at lines 381-387 in to two sentences. A natural break is after reference 51.

 Break the following paragraph up in to shorter sentence too! (Lines 388-392).  Sentence can also be shortened.

Line 403 rather than the word ‘enclosed’, try “older adults should have access to healthy, diverse….. “

Limitation and strengths

Great to see this section included!

Line 434 delete  “to the results obtained”. Your study can be an important contribution!

Line 436 mitigate NEGATIVE effects? (You don’t want to mitigate positive effects!)

Line 437 In other WORDS, (not other hands) again sorry about confusing English idioms.

Conclusion

 Line 451 – enhanced means  to intensify or increase, but usually in a good way.  For eg “You enhance you reputation by doing beneficial, charitable work” .  I would suggest at this line IMAPACTED  “ PIL may be impacted by adversities…”

I encourage you to keep going with revisions!

Author Response

Thanks for the opportunity to review this revision. The paper is much better but there is still a little way to go. below I give detailed comments that I hope will assist, especially in English expression.

Dear reviewer 2, the authors are genuinely grateful for all your precious time and suggestions done to the manuscript. It is amazing to see the reviewer's efforts to build, with us, a stronger document better written and supported.

Abstract

This abstract reads much better. I think avoiding ‘meaning of life’ was a wise choice.  It avoids the connotations of existential experience. It also relates to the references in a meaningful way.

The authors thank you for the comment.

In colloquial English, the article ‘the’ can often be dropped to create an easier flow of written language. I think you can do a minor edit and where not necessary remove THE from ‘purpose in life’.  eg the opening sentence The purpose in life can be more simply put as  “Purpose in life in a complex psychological construct…”. Likewise at line 43 Page 1 Line 50 page 2 and so on. See also line

 Thank you for the suggestions and for the examples, we have reworked the text

Line 50 ‘Better purpose in life’ –I feel ‘better’ is the wrong adjective. It would be ‘stronger  purpose’ or a ‘stronger sense of purpose in life’.  I fell you mean magnitude of purpose here.

Likewise,  if you use stronger purpose, at line 58 you can have a weaker sense of purpose. I feel this now puts purpose onto a continuum of weaker to stronger and you nicely illustrate the variables that affects purpose.  

 Thank you for the suggestions and for the examples, we have reworked the text

Line 64: delete the superfluous THE and impact will be plural: Increasing age negatively impacts purpose in life .. or “Increasing age has been found in some studies to negatively impact purpose in life due to loss of social roles as a consequence of the ageing process.”   The second part fo that sentence does not really make sense. … force the individual to adapt to and leave…  if the individual has adapted to changes brought about by the ageing process surely , they would not be relinquishing these roles?

Thank you for your comment, we do agree, and we have removed the second sentence.

 Line 75 delete the THE. Just ‘Retirement can ….” Line 79 delete the THE in ‘the roles”

 Thank you for your suggestion. We have deleted the “THE”.

Line 80 perhaps consider changing “…experience unpleasant feelings and difficulties in organizing their new life, mainly when this transition occurs abruptly” to ““…experience unpleasant feelings and difficulties in organizing their new life, which is exacerbated when this transition occurs abruptly”.

 We value the insights you shared, and with that, we fixed the error.

Line 84 These findings (plural)

Thank you for the correction that we have done in the text. 

Lines 84-85 Change “low socioeconomic status and dissatisfied jobs” to “…low socioeconomic status and experienced dissatisfaction with their job…”

 Thank you for the correction that we have done in the text. 

Line 88 delete the THE in the working adults.

 Thank you for your suggestion. We have deleted the “THE”.

End of introduction delete the additional text at the end “purpose in”

  Thank you for your suggestion. We have done it.

Overall,  the introduction reads much better! These mainly grammatical and stylistic change should further improve it.

  Thank you for the suggestions, the examples, and all the help provided.

Methods

 You are going to laugh. The opening sentence is one example where the article THE is quite appropriate. “The study design was cross sectional.” I am sorry English is such an absurd language!

 Thank you for your suggestion. We have included the “The”!

2.2 I assume all participants CONSENTED. Change from ‘agreed’ to ‘consented’.

Thank you for your suggestion. We have done it in the text.

Try “A dementia diagnosis was the sole exclusion criterion.” It would be good to say why. Is for example dementia often diagnosed later in the disease meaning that abstract concepts presented in the instruments considered too hard for a people with dementia.  The reason I suggest this is that there is growing literature that questions the exclusion for older people with dementia in research.

 Thank you for your suggestion. We have done it in the text.

Lines 107-109 are a little convoluted! Change “…the large sample of 1330 participants including people from different regions of the country can be consider relevant once it will represents sociodemographic characteristics that are quite illustrative of the population under study.

To “ …the large sample of 1330 participants includes people from different regions of the country and represents sociodemographic characteristics that are illustrative of the population under study.”

Thank you for your suggestion. We have done it in the text.

2.3, Line 113 Instruments were administered or used, not considered.

 We value the insights you shared, and with that, we fixed the error.

Line 144-145

 Change from “Each of these domains is composed of facets of quality of life that summarize the particular domain of quality of life in which they are inserted.” To something more like “each of these domains are characterised by between <lower number> and <higher number> characteristics representing that domain.” Falta mudar isto

 Your comment has been noted and appreciated, we have improved it and explained it better.

And at Line 146 and 147, perhaps “Domain scores are summed to create a global score of overall quality of  life” ( f this is what is done.)

 Thank you for your suggestion. We have done it in the text.

Line 169 “in order to apply preferences, regular cognitive patterns and unconscious or deliberate responses” is it  regular cognitive pattern IN unconscious  or deliberate, or is really and AND?

 We value the insights you shared and fixed the mistake.

 Line 169 delete the ARE at the beginning of the sentence and check that the meaning is accurate. It may be better to say “For example, the systematic style is illustrated by the  statement “I usually do a orderly search… etc”

 Thank you for the suggestions and for the examples, we have reworked the text

Line 175  “Instruments were administered (not applied) by professionals…” you need to say how these organisations were recruited. Did you advertise? Was it through personal contacts?  Were these 19 organisations networked in some way?

Thank you for the suggestions, we have improved it and explained it better.

This detail is very important to understand. The data gathering statement is much better now ( line 181-185).

The authors thank you for the comment.

Line 192 delete the THE in “The multi linear regression”

 Thank you for your suggestion. We have deleted the “The”!

Line 198. You can just say what you did and support  your methods with a reference. Eg change “…assumption of independence of errors was validated with the Durbin-Watson statistic (d=1.59), as suggested by the study [44]”. To  “…assumption of independence of errors was validated using the Durbin-Watson statistic (d=1.59) [44].”

 Thank you for your suggestion. We have done it in the text.

Line 199 “…to identify multicollinearity among the variables, which did not occur.” Which do. Not occur is a result and should move the results section.

Thank you for your suggestion. We move this information to “Results”.

 Line 236 delete ‘te’ ( typo). ‘The’ is not needed here.

 Thank you for your suggestion. We have done it in the text.

Please correct typos at lines 243-244 and formatting at Line 246

Thank you for your suggestion. We have done it in the text.

Line 241 delete ‘the’ in the working individuals.

  Thank you for your suggestion. We have deleted the “The”!

Please correct formatting problems at lines 277-278 and 281-282.

Thank you for your suggestion. We have done it in the text.

Discussion

Great to read the summary of the purpose of the study in the first paragraph!

The authors thank you for the comment.

Please review this paragraph: “It is worth highlighting that, in this study, age is a negative predictor of PIL, which is somehow aligned with the available evidence, which indicates a deterioration of the purpose with increasing age [2,19,22, 46, 47] even though participants still working [46].  Those who still working etc…”

This does not make sense.  Can you please review for meaning and expression?

Perhaps consider writing “…even in participants are still working” the word ‘though’ is maybe not the best choice.

 Thank you for the suggestions and for the examples, we have reworked the text

Be careful of statements like “ it is well known that…” this is supposed to be science! We can talk about assumptions – as assumptions – but we as scientists TEST assumptions!  I would review paragraph at 338 – 342. You make sound clinical sense in helping older people to remain active and socially engaged is but carefully relate this to your results.

Thank you for the suggestion. We have deleted the sentence in order to be more oriented to the study results.

Line 348 be careful again of assumptions. Yes, when retired people don’t meet and engage with colleagues, but they may meet and engage in other ways.  Make sure you relate these comments to your results and  contrast this with the current literature.

Thank you for the suggestion. We have deleted the sentence in order to be more oriented to the study results 

Line 381 change  “It is important to enhance that several studies…”  to perhaps just “ Several studies have shown…” ( use of the word ‘enhance is incorrect.)

 Thank you for your suggestion. We have done it in the text.

Break paragraph at lines 381-387 in to two sentences. A natural break is after reference 51.

 Thank you for your suggestion. We have done it in the text.

 Break the following paragraph up in to shorter sentence too! (Lines 388-392).  Sentence can also be shortened.

 Thank you for your suggestion. We have done it in the text.

Line 403 rather than the word ‘enclosed’, try “older adults should have access to healthy, diverse….. “

 We value the insights you shared and fixed the mistake.

Limitation and strengths

Great to see this section included!

The authors thank you for the comment.

Line 434 delete  “to the results obtained”. Your study can be an important contribution!

  Thank you for your suggestion. We have done it in the text.

Line 436 mitigate NEGATIVE effects? (You don’t want to mitigate positive effects!)

  We value the insights you shared and fixed the mistake. Thank you for your comment, we do agree, and we have included the word “negative”.

Line 437 In other WORDS, (not other hands) again sorry about confusing English idioms.

We thank you for all your important work. We have corrected the mistake.

Conclusion

 Line 451 – enhanced means  to intensify or increase, but usually in a good way.  For eg “You enhance you reputation by doing beneficial, charitable work” .  I would suggest at this line IMAPACTED  “ PIL may be impacted by adversities…”

  We value the insights you shared and fixed the mistake.

I encourage you to keep going with revisions!

Thank you for all your support and help. It was great to have this opportunity to grow in this area with you.